# Incorporating Token Usage into Prompting Strategy Evaluation

## Abstract

In recent years, large language models have demonstrated remarkable performance across diverse tasks. However, their task effectiveness is heavily dependent on the prompting strategy used to elicit output, which can vary widely in both performance and token usage. While task performance is often used to determine prompting strategy success, we argue that efficiency—balancing performance and token usage—can be a more practical metric for real-world utility. To enable this, we propose Big-$O_{tok}$, a theoretical framework for describing the token usage growth of prompting strategies, and analyze Token Cost, an empirical measure of tokens per performance. We apply these to several common prompting strategies to demonstrate their utility and observe that increased token usage leads to drastically diminishing performance returns. Our results validate the Big-$O_{tok}$ and Token Cost analyses and reinforce the need for efficiency-aware evaluations.

## 1 Introduction

Large language models (LLMs) are primarily interacted with through natural language prompts. The composition of a prompt exercises significant, often unexpected, influence over the generated output. This has sparked research into "prompt engineering," the study of prompt design to extract maximum performance from LLMs (White et al., 2023). There are many ways to approach prompt engineering; in this paper, we focus on formalized **prompting strategies**, the overarching paradigms of prompt design (e.g., providing examples of question-answer pairs (Brown et al., 2020)).

As prompting strategies have developed alongside LLMs, benchmark accuracy has emerged as the primary metric for success. New prompting strategies are often tested alongside prior ones on a selection of benchmarks and LLMs, using the gain in accuracy over existing strategies as validation of the new approach. Token usage, if included, is often analyzed post hoc, indicating that it was only a secondary consideration during development. Optimization for performance without regard for token usage can lead to inefficient prompting strategies. Our purpose in this work is to demonstrate another, more holistic approach to prompting strategy evaluation and analysis by (1) proposing Big-$O_{tok}$, a framework for comparing theoretical token usage between distinct prompting strategies and (2) introducing Token Cost (TC), a simple empirical metric to quantify prompting strategy efficiency.

To achieve these goals, we approach prompting strategy efficiency on two fronts: theoretical and empirical. For our theoretical analysis, we derive Big-$O_{tok}$ token complexities for a selection of prompting strategies, similar to time complexity analyses common in software engineering. We substantiate our Big-$O_{tok}$ analyses by evaluating our selection of prompting strategies against common benchmarks using multiple models. We analyze the results of those experiments in terms of TC, to compare how performance and token usage interact. For our experiments, we observe that, while there is performance improvement to be gained from more complex, token-hungry prompting strategies, increasing token usage results in drastically diminishing performance returns.

## 2 Related Work

Although LLM efficiency has been an active area of research for years (see Wan et al. (2024)), underwhelming emphasis has been placed on the efficiency of prompting strategies. Techniques have emerged to reduce token usage, such as frameworks that dynamically manage token usage at inference time—e.g., FrugalGPT (Chen et al., 2023) and LLMLingua (Jiang et al., 2023)—and prompt compression—e.g., Mu et al. (2023). These approaches seek to enable efficient LLM usage in spite of inefficient prompting strategies, whereas this work promotes a focus on prompting strategy efficiency.

Some work, such as Sivarajkumar et al. (2024); Kim et al. (2023); Chu et al. (2024), has been done to evaluate prompting strategies in specific domains, but token usage statistics are often not included. Prompting strategy evaluation is largely left to new prompting strategy proposals (e.g., CoT (Wei et al., 2022)) or benchmarks (e.g., BBH (Suzgun et al., 2023)). These evaluations tend to prioritize benchmark accuracy without significant consideration of token usage. In this work, we demonstrate the relevance of an increased focus on token efficiency and how to proactively incorporate it into prompting strategy analysis.

Some metrics pertaining to cost and efficiency have emerged but tend to be tailored to specific use cases. Wang et al. (2024) proposes a budget-limited evaluation framework to compare prompting strategies and explores why certain prompting strategies may not scale performance with increased compute budget, using raw token and query counts as cost metrics. Wan et al. (2025) uses the weighted average of piecewise accuracy and cost functions to quantify efficiency specifically for self-consistent methods. We propose TC as a simple, independent metric of prompting strategy efficiency and Big-$O_{tok}$ as an analysis that can be used to compare arbitrary prompting strategies without execution.

Some recent prompting strategies, such as Constrained-CoT (Nayab et al., 2024), Concise-CoT (Renze & Guven, 2024), and Algorithm-of-Thoughts (Sel et al., 2024), are designed as optimizations over extant strategies with token usage reduction as a primary motivation. Our hope for this work is that it will enable future prompting strategy development and analysis that similarly prioritizes efficiency.

## 3 Methodology

We explore the importance of token usage both theoretically and empirically. Due to the popularity of LLMs, there exists an infeasible number of possible evaluation combinations[1]. To focus the scope of this paper on token usage, we restrict the number of prompting strategies, benchmarks, and models we use. We discuss our selection processes in Sections 3.1 and 3.2.

### 3.1 Theoretical Analysis

Table 1: Big-$O_{tok}$ token complexities for each prompting strategy.

| Prompting Strategy | Big-$O_{tok}$ | Variables |
|---|---|---|
| Vanilla IO | $O(1)$ | |
| Zeroshot CoT (Kojima et al., 2022) | $O(1)$ | |
| Vanilla Fewshot (Brown et al., 2020) | $O(k)$ | $k$: k-shot exemplars |
| Fewshot CoT (Wei et al., 2022) | $O(k)$ | $k$: k-shot exemplars |
| CoT-SC (Wang et al., 2023b) | $O(pk)$ | $k$: k-shot exemplars; $p$: sampled chains |

We categorize prompting strategies into three broad groups: **(1) linguistic prompt engineering**, which relies on specific phrasing techniques—e.g., Plan-and-Solve (Wang et al., 2023a) or Zeroshot CoT (Kojima et al., 2022); **(2) in-context learning**, which consists of providing examples of task-response pairs before providing the task to the LLM—e.g., Vanilla Fewshot (Brown et al., 2020) or Fewshot CoT (Wei et al., 2022); and **(3) multi-hop**, which is characterized by multiple LLM calls—e.g., Least-To-Most (Zhou et al.,

---

[1]Prompting strategies: 40+ (Vatsal & Dubey, 2024; Chu et al., 2024); Benchmarks: 130+ (Gao et al., 2023); Open-source, benchmarked LLMs: 3200+, as of January, 2025 (Fourrier et al., 2024).

2023), Tree-of-Thought (Yao et al., 2023), or CoT Self-Consistency[2] (Wang et al., 2023b). These three prompting strategy categories roughly correspond to the following Big-$O_{tok}$ complexity classes, respectively: **(1)** constant—e.g., the consistent overhead of "Think step by step" (Wei et al., 2022); **(2)** linear—e.g., the number of fewshot exemplars; and **(3)** polynomial or higher—e.g., the number of multi-hop steps times the number of exemplars. These Big-$O_{tok}$ complexity classes are reflected in Table 1.

To ensure our investigation represents all three categorizations, we select prompting strategies from each. Namely, we choose Vanilla IO (i.e., simply providing the benchmark question) as a baseline; Zeroshot CoT to represent **(1)**; Vanilla Fewshot and Fewshot CoT for **(2)**; and CoT-SC for **(3)**. These strategies are widely adopted, tend to build on each other without significant changes to prompt design, and demonstrate an organic evolution of prompting strategies over several years (Chu et al., 2024).

The purpose of Big-$O_{tok}$ is to provide an objective representation of the theoretical token usage growth rate of a given prompting strategy, enabling direct comparison with the Big-$O_{tok}$ of other prompting strategies. Big-$O_{tok}$ is based on Big-O notation (Knuth, 1976) and thus we rely on the terminology and definitions associated with it.

Big-$O_{tok}$ describes the asymptotic growth of token usage as a function of variables[3] in the prompting strategy (e.g., the number of fewshot examples). It is derived analogously to Big-O time complexity: by considering how token usage increases as prompting strategy variables approach infinity. The variables with the highest growth rate dominate the other terms (e.g., constants, lower-order variables, and scalars), which can then be omitted. Big-$O_{tok}$ token complexity can often be derived from a natural language description of a prompting strategy. We provide sample derivations for the Big-$O_{tok}$ functions from Table 1 in Appendix A.

### 3.2 Empirical Analysis

We test our selection of prompting strategies against three common benchmarks using three LLMs. To perform the empirical evaluations, we leverage LM Evaluation Harness (Biderman et al., 2024; Gao et al., 2023), a framework aimed at increasing the reproducibility of LLM evaluations.

We base our selection of models on recency, popularity, and size. We do not use commercial models due to budget constraints[4]. We select Llama 3.1 8B Instruct (Dubey et al., 2024), Qwen 2.5 14B Instruct, and Qwen 2.5 32B Instruct (Qwen et al., 2025). This selection provides coverage of various sizes of smaller models (each approximately doubling the parameter count of the prior) and diversity of origin, to ensure multiple approaches to data collection, training, and alignment are represented[5].

For benchmarks, we select BBH (Suzgun et al., 2023), GSM8K (Cobbe et al., 2021), and MMLU (Hendrycks et al., 2021). This represents a diverse group of general-purpose benchmarks based on typical accuracy ranges[6] and response type[7]. For fewshot prompting strategies, we use 3 exemplars for BBH, 8 for GSM8K, and 4 for MMLU[8].

## 4 Results

### 4.1 Big-$O_{tok}$

To substantiate our Big-$O_{tok}$ analyses, we use the observed token usages from our experiments to calculate the relative token usage ratios between prompting strategies. We derive theoretical estimates of those ratios from our Big-$O_{tok}$ functions by substituting in the values from our experiments for the variables in Big-$O_{tok}$

---

[2]Abbreviated as CoT-SC$_n$, where $n$ is the number of sampled chains.

[3]We treat the initial input that the prompting strategy modifies (e.g., a benchmark question) to be constant and exclude it for simplicity. Similarly, we treat additive adjustments to the input or output (e.g., CoT's "Think step by step" (Wei et al., 2022)) as constants.

[4]See Appendix C.3 for cost estimates for commercial APIs.

[5]We choose two Qwen 2.5 models to facilitate a comparison between model size, found in Appendix D.2.

[6]GSM8K: 80-95%; BBH: 50-87%; MMLU: 70-92% (Fourrier et al., 2024; Dubey et al., 2024; Qwen et al., 2025).

[7]GSM8K: free response number; BBH: free response text; MMLU: multiple choice.

[8]These numbers are based on the availability of CoT examples in LM Evaluation Harness and closely reflect the number of examples suggested in CoT-SC (Wang et al., 2023b).

Table 2: Theoretical and observed token usage ratios between prompting strategies, averaged over the three benchmarks. Values are formatted as *<theoretical>; <observed>* and derived by $\frac{num\ tokens_2}{num\ tokens_1}$, where $num\ tokens_2 > num\ tokens_1$.

| **Column:** Higher Token Usage (Numerator) **Row:** Lower Token Usage (Denominator) | CoT-SC$_{10}$ | CoT-SC$_5$ | Fewshot CoT | Vanilla Fewshot | Zeroshot CoT |
|---|---|---|---|---|---|
| Vanilla IO | 50; 29.3 | 25; 14.6 | 5; 3.0 | 5; 2.2 | 1; 1.3 |
| Zeroshot CoT | 50; 23.4 | 25; 11.7 | 5; 2.4 | 5; 1.7 | |
| Vanilla Fewshot | 10; 13.5 | 5; 6.8 | 1; 1.4 | | |
| Fewshot CoT | 10; 9.9 | 5; 5.0 | | | |
| CoT-SC$_5$ | 2; 2.0 | | | | |

(e.g., $p = 5$ and $\bar{k} = 5$ for CoT-SC$_5$). The results of that comparison are found in Table 2. We expect noise in the observed token usage due to: inherent token usage (e.g., chat templates); the relatively low values of prompting strategy variables (e.g., $k = 3$ fewshot exemplars for BBH); and the unpredictability of LLM output. However, while the observed and theoretical factors are not perfect matches, our findings do correctly align with the Big-O$_{\text{tok}}$ token complexity classes discussed in Section 3.1 (constant, linear, and polynomial). In other words,

$$T_{\text{O}_{\text{tok}}(1),\text{b,m}}() < T_{\text{O}_{\text{tok}}(k),\text{b,m}}(k) < T_{\text{O}_{\text{tok}}(pk),\text{b,m}}(p, k)$$

where $T_{\text{O}_{\text{tok}},\text{b,m}}(x)$ is the observed token usage for each prompting strategy in the O$_{\text{tok}}$ complexity class, benchmark $b$, model $m$, and variable $x$ in our experiments.

## 4.2 Token Cost

To quantify token efficiency, we discuss the results from our experiments in terms of Token Cost (TC). We define TC as the number of tokens[9] per percentage point of accuracy (expressed as $\frac{t}{p}$). The inverse of TC can be thought of as token efficiency; thus, relatively high TC is less efficient while lower TC is more efficient. We use this metric, $\frac{t}{p}$, rather than the inverse, $\frac{p}{t}$, because we find it to be more intuitive in the context of prompting strategies. We include an expanded discussion on interpreting TC, including edge cases, in Appendix B.

The results of the experiments outlined in Section 3.2 are found in Figure 1. Across all benchmarks and models, our empirical results follow consistent trend lines (of the form $y = \log(\log(x))$), reflecting the diminishing accuracy returns from increased token usage. In other words, it requires significantly more tokens to realize accuracy gains as token usage increases. To discuss this trend in terms of TC, we explore both *average* and *marginal* TC[10]. Average TC is simply the token usage divided by the accuracy for a given prompting strategy ($\frac{num\ tokens_{obs}}{accuracy_{obs}}$). Marginal TC is the change in token usage to realize the change in accuracy between two prompting strategies. In other words, for $num\ tokens_2 >= num\ tokens_1$,

$$\frac{num\ tokens_2 - num\ tokens_1}{accuracy_2 - accuracy_1}$$

Both average and marginal TC can be thought of as the slope between two points (one being the origin, for average TC), which represents the expected cost, in tokens, of adding one point of accuracy.

Across all experiments, the average TC for the prompting strategy with the lowest accuracy is 5.0 $\frac{t}{p}$, while that of the highest performing prompting strategy is 119.4 $\frac{t}{p}$, a more than 20x increase in average TC and,

---

[9]Token counts are estimated by $\frac{num\ characters}{4}$.

[10]We use the average of ratios when discussing each to give equal weighting to every observation.

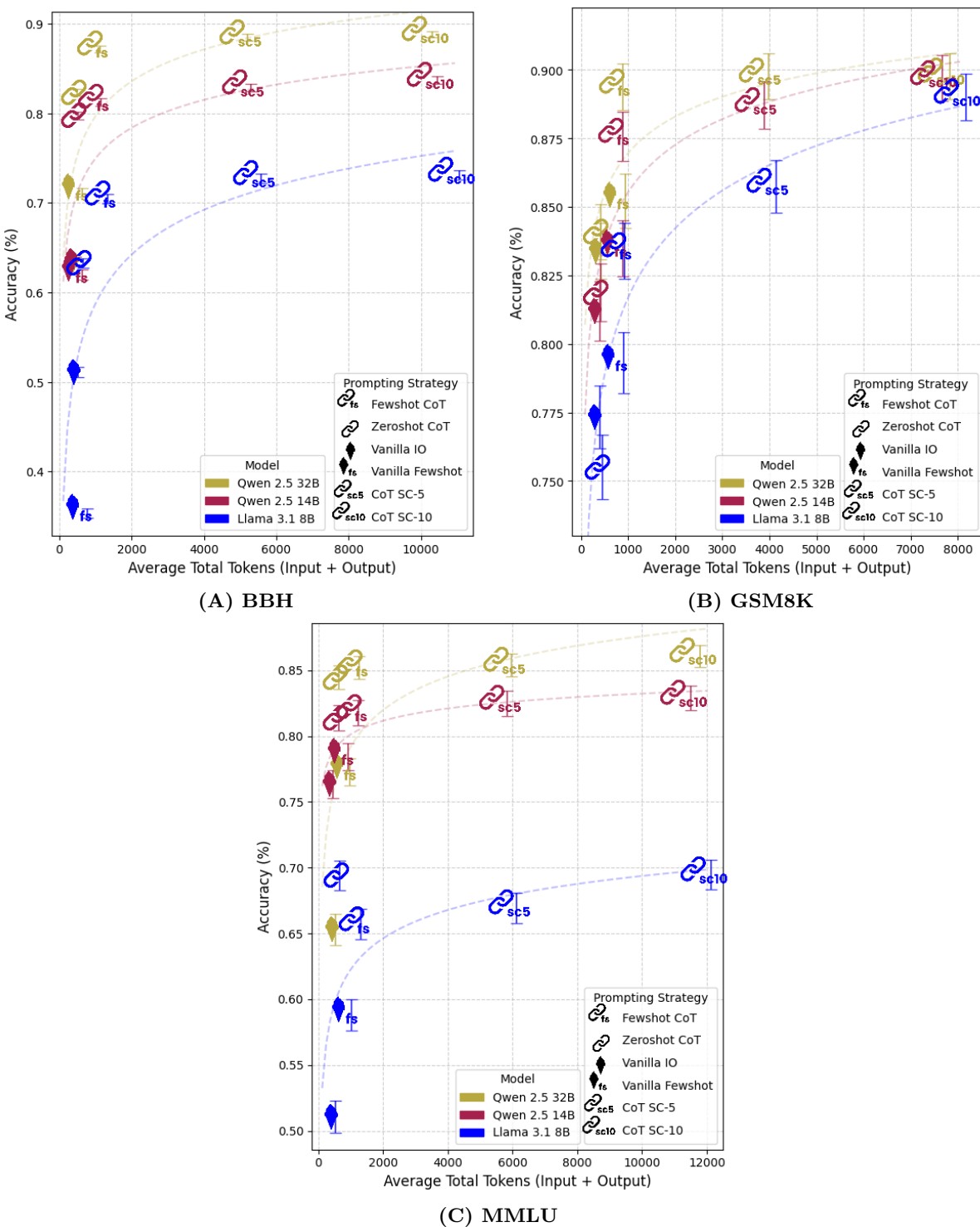

Figure 1: Accuracy vs. token usage plots with standard error bars for various prompting strategies, models, and benchmarks. The trend lines reflect the rapid growth of TC for these strategies.

inversely, 20x *decrease* in efficiency. This is reflected in the plots in Figure 1 in that even the worst performing prompting strategies still attain relatively high accuracy and more complex ones make only small gains over

that for vastly more token usage. Using accuracy as the sole metric ignores the drastic decrease in efficiency that can result from increased token usage.

In Figure 1, we observe an initially steep curve, indicating that tokens are traded relatively efficiently for accuracy, followed by a plateau, where tokens are traded more inefficiently for accuracy. To compare the marginal TC along this curve, we select Vanilla IO, Fewshot CoT, and CoT-SC$_{10}$ which tend to lie on the extremes of the trend lines. Across all experiments, the marginal TC between Vanilla IO and Fewshot CoT is 65.3 $\frac{t}{p}$ while the marginal TC between Fewshot CoT and CoT-SC is 6701.8 $\frac{t}{p}$, a decrease in efficiency of more than two orders of magnitude. In a high-stakes scenario where accuracy is paramount, pursuing performance gains at a rate of 6701.8 $\frac{t}{p}$ may be an acceptable cost. For many real-world scenarios, increasing accuracy at a rate of 65.3 $\frac{t}{p}$ might be more reasonable. TC provides an intuitive way to compare the tradeoff between token usage and performance and allows for more informed prompting strategy and variable selection.

We perform an ablation study, found in Appendix D.1, on the number of fewshot exemplars. We observe that incrementally increasing the number of fewshot examples follows a similar trend of diminishing returns as token usage increases.

All information for reproducing our results, as well as our verbatim results, are detailed in Appendices C.4 and C.5.

## 5 Conclusion

Token usage represents a significant, yet often underrepresented, component of prompting strategy evaluation. To facilitate the comparison of token usage between distinct prompting strategies, we present Big-O$_{\text{tok}}$ token complexity and substantiate it empirically by comparing predicted token usages to those derived from our experiments. We analyze our experiments in terms of Token Cost and use it to demonstrate the tradeoffs between token usage and performance. Our analyses demonstrate the importance of including token usage in prompting strategy evaluation and validate Big-O$_{\text{tok}}$ and Token Cost as viable means of doing so.

### Limitations

To focus the contribution of this work, we make a number of thoughtful concessions, such as the models, benchmarks, and prompting strategies we use. We explicitly justify the most relevant limitations in the main text above–such as in Sections 3.1 and 3.2–and note other minor limitations here to further demonstrate the purpose and scope of this work. Despite the limitations, we maintain that the core premise of this work–demonstrating how to incorporate token usage into prompting strategy evaluation–remains broadly applicable beyond our experiments.

**Prompting Strategies.** In Section 1, we narrow the scope of this work to a subset of the broader prompt engineering landscape: formalized prompting strategies. As detailed throughout the Appendix, we strive to control for factors extraneous to the minimal instantiation of each prompting strategy. We do so to isolate the effects of token usage dictated by the prompting strategies and the resultant benchmark performance to demonstrate how a significant area of research has become prone to inefficiencies due to a lack of relevant metrics. Although we focus on formalized prompting strategies, Big-O$_{\text{tok}}$ and TC are useful metrics for other aspects of prompt engineering as well, such as linguistic and language choices.

As noted in Sections 1 and 2, the focus of this work is on incorporating token usage into prompting strategy evaluation and analysis. We do not aim to solve issues of prompting strategy efficiency but instead provide methods for quantifying it, both for researchers and practitioners. To maintain that scope, we do not explore nor propose specific methods of optimizing prompts and instead focus on introducing insightful metrics and demonstrating their utility in practice.

To do so, we approximate tokens as $\frac{\text{num characters}}{4}$. The reason we use this approximation instead of the actual token counts is because tokenizer- and model-specific idiosyncrasies can result in numerous valid tokenizations of the same text. For our general-purpose benchmarks in English, it is a well-established

approximation[11]. However, we recognize that this approximation may not apply to all models, languages, and domains.

The results we present here represent a single thread of prompting strategy evolution; we expect the results to follow a predictable pattern (e.g., diminishing accuracy returns) because there are no drastic changes to the principles underlying our selection of prompting strategies. It is likely that fundamentally different prompting strategies, such as Least-to-Most (Zhou et al., 2023) or Algorithm of Thoughts (Sel et al., 2024), would not follow our observed trend lines and thus we do not make any generalized claims based on our observations.

**Models.** Depending on the data collection and processing methods used by LLM creators, benchmark data leakage could influence our empirical results, as demonstrated by Mirzadeh et al. (2025). LLMs that were trained on data that included BBH question-answer pairs, for example, could be influenced by prompting strategies to a lesser degree than those that were not. We use models from multiple sources in conjunction with multiple benchmarks to mitigate the potential effects of data leakage.

In this work, we exclusively consider autoregressive, text-to-text ("traditional") LLMs because most prompting strategies are optimized for them. We recognize, however, that multimodal and, more recently, reasoning LLMs have become increasingly relevant (DeepSeek-AI et al., 2025; Yin et al., 2024; Caffagni et al., 2024). We exclude them from our investigation here for a number of reasons: (1) prompt engineering specific to such models is a nascent field and distinct from prompt engineering for traditional LLMs[12] Wu et al. (2024); (2) due to the recency of reasoning models, there are very few (especially open-source) models available; and (3) the inclusion of multimodal benchmarks and the use of reasoning models would drastically increase the compute required to undertake a similar study. Nevertheless, we maintain that the efficiency-aware metrics explored here remain relevant to such models since they function simply on tokenized inputs and outputs. We see prompting strategies designed for multimodal and, particularly, reasoning LLMs as a significant avenue for future research and are hopeful that Big-$O_{\text{tok}}$ and TC will be incorporated into their development.

While we believe it a fair comparison that lends itself to real-world deployment, we recognize that running our selection of prompting strategies with relatively small LLMs on a subset of benchmarks does not fully reflect the performance of the strategies under all conditions. It is very likely, for example, that the CoT prompting strategy (Wei et al., 2022) would be leveraged better by Llama 3.1 405B than by Llama 3.1 8B, due to the former exhibiting superior reasoning capabilities (Dubey et al., 2024). The strength of an LLM may magnify the disparity between a specific prompting strategy and others.

**Benchmarks.** Similarly, while we attempt to cover a breadth of domains in our selection of benchmarks, this selection may fail to highlight the strengths of some prompting strategies over others in certain domains. Our purpose is not to rank prompting strategies but to analytically explore the tradeoffs between token usage and benchmark accuracy. We recognize, however, that our selection of benchmarks may fail to cover the strength of a particular prompting strategy entirely, which may paint it in a worse light than it deserves. To mitigate this point, we focus on generalist prompting strategies and benchmarks and detail our selection processes in Sections 3.1 and 3.2.

A common issue in LLM benchmarking is reliable answer extraction. Often, regular expressions are used, which are not robust to the unpredictable output formats an LLM may generate. We rely largely on the extraction methods from LM Evaluation Harness but observe certain inaccuracies in answer extraction. This is an open problem in LLM research (Yu et al., 2025). For this project, we rely on consistency in answer extraction methods between experiments but recognize that certain correct answers may be marked incorrectly.

---

[11]Common commercial LLM providers also suggest this estimate despite models trained in different years, across multiple languages, and on diverse domains (e.g., Google: `https://ai.google.dev/gemini-api/docs/tokens`; OpenAI: `https://platform.openai.com/tokenizer`).

[12]For reasoning models, some commercial providers even advise against the use of established prompting strategies (see `https://platform.openai.com/docs/guides/reasoning-best-practices`).

**Big-O$_{tok}$.** A minor limitation of Big-O$_{tok}$ is a lack of differentiation between input and output tokens. It is inherently more expensive for an LLM to generate output tokens than to process input tokens due to its autoregressive nature, a fact that is reflected in the pricing structure of common commercial APIs[13]. We considered that differentiating between input and output tokens would have introduced excessive complexity to Big-O$_{tok}$, particularly since it does not serve as a precise measure. We instead consider combined token usage (input **and** output) to provide a holistic view of token consumption.

Another limitation of Big-O$_{tok}$ is the decreased range of values for prompting strategy variables. While variables in traditional Big-O analyses often span many orders of magnitude, variables in prompting strategies tend to be lower (typically $<= 100$) (Brown et al., 2020; Wang et al., 2023b). As noted in Section 4, the relatively low values for those variables could result in extraneous factors (e.g., chat templates, model idiosyncrasies, etc.) limiting their impact on overall token usage. However, we observe that, even for extremely low values (e.g., $k = 3$ fewshot examples for BBH), the token usages from each experiment align with the expected Big-O$_{tok}$ token complexity classes. Although not a precise measure, Big-O$_{tok}$ can still provide useful insights for expected prompting strategy token usage, even for small values.

### Broader Impact Statement

LLM usage incurs real-world monetary and environmental costs (Schwartz et al., 2020; Dhar, 2020; Wu et al., 2022). This work promotes the consideration of token usage in prompting strategy development and evaluation to increase the long-term efficiency of LLM inference.

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

Table 3: The theoretical token complexities for various prompting strategies. Greek letters represent the overhead associated with an IO pair (assumed constant per prompting strategy) and Roman letters represent variables.

| Prompting Strategy | Big-$O_{tok}$ | Token Complexity | Variables | Values[†] |
|---|---|---|---|---|
| MVIO | $O(1)$ | 1 | | |
| Vanilla IO | $O(1)$ | $1 + \psi$ | | |
| Zeroshot CoT (Kojima et al., 2022) | $O(1)$ | $1 + \alpha$ | | |
| Vanilla Fewshot (Brown et al., 2020) | $O(k)$ | $1 + k$ | $k$: k-shot exemplars | $k = 0, 1, 10 - 100$ |
| Fewshot CoT (Wei et al., 2022) | $O(k)$ | $1 + \alpha + k + k\alpha$ | $k$: k-shot exemplars | $k = 8$ |
| CoT-SC (Wang et al., 2023b) | $O(pk)$ | $p(1 + \alpha + k + k\alpha)$ | $k$: k-shot exemplars; $p$: sampled chains | $k = 4 - 8$; $p = 40$ |

[†] Values suggested in the papers that originally introduced the prompting strategy.

Table 4: The token usage growth rate over MVIO per prompting strategy, derived from Table 1 using the variables used in our experiments.

| Prompting Strategy | Token Usage Growth Rate | | |
|---|---|---|---|
| | BBH | GSM8K | MMLU |
| Vanilla IO | 1 | 1 | 1 |
| CoT Zeroshot | 1 | 1 | 1 |
| Vanilla Fewshot | 3 | 8 | 4 |
| Fewshot CoT | 3 | 8 | 4 |
| CoT-SC$_5$ | 15 | 40 | 20 |
| CoT-SC$_{10}$ | 30 | 80 | 40 |

Shukang Yin, Chaoyou Fu, Sirui Zhao, Ke Li, Xing Sun, Tong Xu, and Enhong Chen. A survey on multimodal large language models. *National Science Review*, 11(12):nwae403, 11 2024. ISSN 2095-5138. doi: 10.1093/nsr/nwae403. URL https://doi.org/10.1093/nsr/nwae403.

Qingchen Yu, Zifan Zheng, Shichao Song, Zhiyu Li, Feiyu Xiong, Bo Tang, and Ding Chen. xfinder: Large language models as automated evaluators for reliable evaluation. In *The Thirteenth International Conference on Learning Representations*, 2025. URL https://openreview.net/forum?id=7UqQJUKaLM.

Denny Zhou, Nathanael Schärli, Le Hou, Jason Wei, Nathan Scales, Xuezhi Wang, Dale Schuurmans, Claire Cui, Olivier Bousquet, Quoc V Le, and Ed H. Chi. Least-to-most prompting enables complex reasoning in large language models. In *The Eleventh International Conference on Learning Representations*, 2023. URL https://openreview.net/forum?id=WZH7099tgfM.

# A  Big-$O_{tok}$ Analysis

We provide an expanded token complexity analysis for each prompting strategy examined in the main text in Table 3. We define the minimally viable IO pair (MVIO) for a given benchmark question to be the full text of the question and the minimum amount of text to convey the answer (e.g., Question: "How many days are there in a week?"; Answer: "7"). All other prompting strategies that incur additive adjustments to the input or output (e.g., the natural language thinking induced by CoT's "Think step by step" (Wei et al., 2022)) are treated as constant overheads on top of the MVIO, which are represented by Greek letters.

Table 4 shows theoretical token usage ratios based on Big-$O_{tok}$. The expected token usage ratios used in Table 2 are based on these, averaged across the three benchmarks.

We include examples of Big-$O_{\text{tok}}$ derivations for each prompting strategy examined in the main text in Figure 2.

## B    Interpreting Token Cost

Although we include a thorough example of using TC for prompting strategy analysis, we seek to provide an expanded discussion on its interpretation here. For brevity, in the main text we state that, generally, low TC can be thought of as more efficient and high TC as less efficient. In most instances, this will hold true; however, there are some plausible edge cases that deserve consideration.

### B.1    Average TC

One such edge case is exploiting extremely low token usages to achieve low average TC. For example, consider a multiple-choice benchmark with four options: (A), (B), (C), and (D). Assuming uniform distribution across the possible answers, a prompting strategy that would yield 25% accuracy (assuming the LLM could consistently produce the correct output) might be: "Output (B)." At approximately 4 combined input and outputs tokens, such a prompting strategy would achieve an average TC of 0.16 $\frac{t}{p}$, a value much lower than any of our observed values in Table 6. This demonstrates the need to test prompting strategies for generalizability.

### B.2    Marginal TC

For marginal TC, there are additional edge cases to consider. The formula for calculating marginal TC,

$$\frac{num\ tokens_2 - num\ tokens_1}{accuracy_2 - accuracy_1}$$

where $num\ tokens_2 >= num\ tokens_1$, allows for negative values. Despite being "low," a negative marginal TC value indicates extreme inefficiency since the prompting strategy will have consumed more tokens to achieve lower accuracy.

In the unlikely event that $num\ tokens_2 == num\ tokens_1$, marginal TC will not provide useful information. The more efficient prompting strategy would, in that case, be the one that attained higher accuracy. Similarly, if $accuracy_2 == accuracy_1$, marginal TC cannot be calculated, but the prompting strategy that consumed fewer tokens can be considered more efficient.

## C    Empirical Evaluation Details

### C.1    Detailed Results

We provide detailed results from our experiments in Table 6. Note that the results presented for each prompting strategy, model, and benchmark combination are from a single execution with the number of samples noted in Table 9.

We removed empty outputs and outputs more than four standard deviations from the mean (e.g., instances where the LLM generated a looping output) from our token usage statistics. Such erroneous outputs were surprisingly common for Llama 3.1 8B Instruct. Details of the number of outputs removed from consideration are detailed in Table 5.

### C.2    Additional Observations

While our experiments were simple, we made a number of interesting observations that may warrant further analysis in future work.

The initial motivation behind fewshot prompting strategies was to define a pattern that the LLM would then follow in its answer (Brown et al., 2020; Wei et al., 2022). For Fewshot CoT, this pattern included a reasoning

**The minimum text required to communicate the question and answer.**

1: Minimally viable text for the question + answer

⋮

**Token Complexity: T() = 1**
**Big-$O_{tok}$: $O_{tok}(1)$**

(A) MVIO

**The minimum text required to communicate the question and the generated output.**

1: Minimally viable text for the question
ψ: LLM-generated output

⋮

**Token Complexity: T() = 1 + ψ**
**Big-$O_{tok}$: $O_{tok}(1)$**

(B) Vanilla IO

**"The core idea of our method is simple...: add *Let's think step by step*, or a a [sic] similar text...to extract step-by-step reasoning."**

α: the "trigger prompt" + the elicited reasoning *(treated as constant overhead)*
1: Minimally viable IO (question + answer)

⋮

**Token Complexity: T() = 1 + α**
**Big-$O_{tok}$: $O_{tok}(1)$**

(C) Zeroshot CoT

**"[F]ew-shot works by giving K examples of context and completion, and then one final example of context, with the model expected to provide the completion."**

1: The question and the answer (assumed MVIO)
k: Number of fewshot examples (in the form of MVIO)

⋮

**Token Complexity: T(k) = 1 + k**
**Big-$O_{tok}$: $O_{tok}(k)$**

(D) Vanilla Fewshot

**"Our proposed approach is to augment each exemplar in few-shot prompting with a chain of thought for an associated answer."**

k: Number of fewshot examples
α: Reasoning overhead
1: Minimally viable IO

⋮

**Token Complexity: T(k) = 1 + α + k + kα**
**Big-$O_{tok}$: $O_{tok}(k)$**

(E) Fewshot CoT

**"We first prompt the language model with chain-of-thought prompting, then…generate a diverse set of reasoning paths."**

CoT $\begin{cases} \text{k: Number of fewshot examples} \\ \text{α: Reasoning overhead} \\ \text{1: Minimally viable IO} \end{cases}$
p: Number of generated reasoning paths

⋮

**Token Complexity: T(p,k) = p(1 + α + k + kα)**
**Big-$O_{tok}$: $O_{tok}(pk)$**

(F) CoT-SC

Figure 2: Sample derivations of Big-$O_{tok}$. The textual descriptions in each figure are drawn from the following sources: (C) Kojima et al. (2022); (D) Brown et al. (2020); (E) Wei et al. (2022); (F) Wang et al. (2023b). Note that for (D), the fewshot examples are equivalent to the MVIO and we make the assumption that the LLM follows that pattern.

Table 5: The percentage of total IO pairs removed from token usage statistics. Pairs were removed if the output was empty or the length of the output was more than 4 standard deviations from the mean.

| Model | Benchmark | Percentage of Total IO Pairs Excluded | | | | | |
|---|---|---|---|---|---|---|---|
| | | Vanilla IO | Vanilla Fewshot | Zeroshot CoT | Fewshot CoT | CoT-SC$_5$ | CoT-SC$_{10}$ |
| Llama 3.1 8B Instruct | BBH | 2.53 | 8.59 | 2.90 | 5.04 | 4.09 | 4.00 |
| | GSM8K | 0.68 | 1.14 | 0.76 | 0.91 | 1.27 | 1.28 |
| | MMLU | 1.18 | 1.76 | 1.89 | 2.35 | 1.89 | 1.93 |
| Qwen 2.5 14B Instruct | BBH | 0.29 | 0.12 | 0.23 | 0.49 | 0.24 | 0.19 |
| | GSM8K | 0.30 | 0.45 | 0.23 | 0.23 | 0.36 | 0.09 |
| | MMLU | 0.07 | 0.65 | 0.20 | 0.46 | 0.20 | 0.38 |
| Qwen 2.5 32B Instruct | BBH | 0.22 | 0.09 | 0.26 | 0.29 | 0.16 | 0.10 |
| | GSM8K | 0.00 | 0.00 | 0.08 | 0.38 | 0.18 | 0.13 |
| | MMLU | 0.20 | 0.46 | 0.07 | 0.07 | 0.10 | 0.11 |

chain that led to the right answer (Wei et al., 2022). Now that LLMs are more capable and often aligned with human preferences after training, their default response to a question is to explain their reasoning before providing a response. This results in high output token usage even for Vanilla IO and Zeroshot CoT. Interestingly, **the reasoning chains (averaged, per benchmark) that the LLMs produced for Zeroshot CoT were longer in every instance than the ones produced for Fewshot CoT**, in some cases more than twice as long. Nonetheless, Fewshot CoT yielded accuracy improvements for nearly every benchmark. This supports the idea that the in-context learning of correct reasoning chains does positively influence the correctness of the generated reasoning chain, even if the LLM's default output is constrained.

That trend does not apply to Vanilla IO and Vanilla Fewshot, however. While Vanilla Fewshot outperforms Vanilla IO in almost every benchmark, the output token usage is less in most instances. This is likely caused by the pattern that is matched during Vanilla Fewshot, where the example outputs are the minimum number of tokens to convey the answer (e.g., for multiple-choice: "(A)").

We also observed how the quality of the fewshot reasoning chains provided as a part of Fewshot CoT affected performance. While Fewshot CoT yielded modest accuracy gains over Zeroshot CoT for BBH (4.6%) and GSM8K (6.3%), it actually registered a 0.9% accuracy *loss* on MMLU. This prompted an investigation into the CoT fewshot exemplars included in LM Evaluation Harness (Gao et al., 2023). The reasoning chains were significantly shorter than for the other two benchmarks. Interestingly, recent work on using concise reasoning chains, such as Constrained-CoT (Nayab et al., 2024) and Concise-CoT (Renze & Guven, 2024), has demonstrated performance improvements from shorter chains of thought. A potentially insightful future work could explore how the form and content of intentionally concise chains of thought influence LLM performance to explain this discrepancy.

### C.3 Cost Estimates for Commercial Models

The cost estimates found in Table 7 are derived from the pricing pages for Anthropic[14] and OpenAI[15], accessed on January 31, 2025. We do not include the effects of prompt caching.

---

[14]https://anthropic.com/pricing#anthropic-api
[15]https://openai.com/api/pricing/

| | | Strategy | Avg. Tokens | | | Acc.[*] | Std. Error | Average TC |
|---|---|---|---|---|---|---|---|---|
| | | | In | Out | Total | | | |
| Llama 3.1 8B Instruct | BBH | Vanilla IO | 172 | 221 | 393 | 51.1 | 0.56 | 7.70 |
| | | Vanilla 3-shot | 420 | 226 | 646 | 35.4 | 0.54 | 18.28 |
| | | Zeroshot CoT | 178 | 351 | 530 | 63.2 | 0.55 | 8.38 |
| | | 3-shot CoT | 876 | 335 | 1212 | 70.5 | 0.51 | 17.20 |
| | | CoT-SC$_5$ | 4378 | 1089 | 5468 | 72.7 | 0.50 | 75.19 |
| | | CoT-SC$_{10}$ | 8758 | 2177 | 10935 | 73.1 | 0.50 | 149.58 |
| | GSM8K | Vanilla IO | 122 | 162 | 284 | 77.3 | 1.15 | 3.68 |
| | | Vanilla 8-shot | 639 | 147 | 787 | 79.3 | 1.12 | 9.93 |
| | | Zeroshot CoT | 126 | 203 | 330 | 75.5 | 1.18 | 4.37 |
| | | 8-shot CoT | 654 | 145 | 800 | 83.4 | 1.02 | 9.60 |
| | | CoT-SC$_5$ | 3275 | 751 | 4026 | 85.7 | 0.96 | 46.96 |
| | | CoT-SC$_{10}$ | 6550 | 1499 | 8050 | 89.0 | 0.86 | 90.44 |
| | MMLU | Vanilla IO | 201 | 201 | 402 | 51.1 | 1.24 | 7.88 |
| | | Vanilla 4-shot | 711 | 208 | 920 | 58.8 | 1.19 | 15.65 |
| | | Zeroshot CoT | 207 | 342 | 550 | 69.4 | 1.13 | 7.93 |
| | | 4-shot CoT | 1008 | 182 | 1191 | 65.7 | 1.16 | 18.13 |
| | | CoT-SC$_5$ | 5037 | 955 | 5993 | 66.9 | 1.16 | 89.53 |
| | | CoT-SC$_{10}$ | 10076 | 1929 | 12006 | 69.5 | 1.12 | 172.76 |
| Qwen 2.5 14B Instruct | BBH | Vanilla IO | 134 | 197 | 332 | 63.5 | 0.51 | 5.24 |
| | | Vanilla 3-shot | 379 | 143 | 523 | 62.0 | 0.55 | 8.44 |
| | | Zeroshot CoT | 140 | 254 | 395 | 79.7 | 0.42 | 4.96 |
| | | 3-shot CoT | 830 | 195 | 1026 | 81.2 | 0.43 | 12.64 |
| | | CoT-SC$_5$ | 4155 | 1011 | 5166 | 82.8 | 0.41 | 62.37 |
| | | CoT-SC$_{10}$ | 8310 | 2028 | 10339 | 83.7 | 0.40 | 123.52 |
| | GSM8K | Vanilla IO | 101 | 180 | 281 | 81.2 | 1.08 | 3.47 |
| | | Vanilla 8-shot | 618 | 150 | 768 | 83.5 | 1.02 | 9.21 |
| | | Zeroshot CoT | 104 | 194 | 299 | 81.9 | 1.06 | 3.65 |
| | | 8-shot CoT | 633 | 125 | 759 | 87.6 | 0.91 | 8.67 |
| | | CoT-SC$_5$ | 3168 | 602 | 3770 | 88.7 | 0.87 | 42.51 |
| | | CoT-SC$_{10}$ | 6337 | 1221 | 7559 | 89.7 | 0.84 | 84.28 |
| | MMLU | Vanilla IO | 162 | 162 | 325 | 76.4 | 1.05 | 4.26 |
| | | Vanilla 4-shot | 673 | 130 | 804 | 78.4 | 1.03 | 10.25 |
| | | Zeroshot CoT | 169 | 347 | 516 | 81.4 | 0.97 | 6.35 |
| | | 4-shot CoT | 965 | 163 | 1129 | 81.8 | 0.96 | 13.81 |
| | | CoT-SC$_5$ | 4829 | 883 | 5713 | 82.5 | 0.95 | 69.26 |
| | | CoT-SC$_{10}$ | 9650 | 1743 | 11394 | 82.9 | 0.94 | 137.47 |
| Qwen 2.5 32B Instruct | BBH | Vanilla IO | 134 | 201 | 336 | 63.4 | 0.50 | 5.30 |
| | | Vanilla 3-shot | 379 | 144 | 523 | 71.2 | 0.49 | 7.36 |
| | | Zeroshot CoT | 141 | 242 | 383 | 82.3 | 0.39 | 4.66 |
| | | 3-shot CoT | 830 | 182 | 1012 | 87.2 | 0.37 | 11.62 |
| | | CoT-SC$_5$ | 4153 | 938 | 5092 | 88.5 | 0.36 | 57.56 |
| | | CoT-SC$_{10}$ | 8308 | 1883 | 10191 | 88.8 | 0.35 | 114.75 |
| | GSM8K | Vanilla IO | 101 | 197 | 298 | 83.4 | 1.02 | 3.58 |
| | | Vanilla 8-shot | 618 | 192 | 810 | 85.2 | 0.98 | 9.51 |
| | | Zeroshot CoT | 104 | 199 | 304 | 84.1 | 1.01 | 3.62 |
| | | 8-shot CoT | 633 | 135 | 769 | 89.4 | 0.85 | 8.60 |
| | | CoT-SC$_5$ | 3168 | 687 | 3856 | 89.8 | 0.83 | 42.96 |
| | | CoT-SC$_{10}$ | 6337 | 1373 | 7711 | 89.8 | 0.83 | 85.90 |
| | MMLU | Vanilla IO | 162 | 249 | 412 | 65.3 | 1.18 | 6.32 |
| | | Vanilla 4-shot | 671 | 194 | 865 | 77.3 | 1.05 | 11.21 |
| | | Zeroshot CoT | 169 | 357 | 526 | 84.5 | 0.89 | 6.23 |
| | | 4-shot CoT | 966 | 186 | 1153 | 85.2 | 0.87 | 13.53 |
| | | CoT-SC$_5$ | 4827 | 1013 | 5840 | 85.4 | 0.87 | 68.37 |
| | | CoT-SC$_{10}$ | 9652 | 2030 | 11682 | 86.1 | 0.86 | 135.70 |

[*] Accuracy as a percentage.

Table 6: Detailed results from the empirical evaluation described in Section 3.2.

| Model | Price$_\text{input}$* | Price$_\text{output}$* | Cost$^\dagger$ (US$) | | |
|---|---|---|---|---|---|
| | | | BBH | GSM8K | MMLU |
| GPT-4o | 2.5 | 10.0 | 488.25 | 72.53 | 121.57 |
| GPT-4o-mini | 0.15 | 0.6 | 29.29 | 4.35 | 7.29 |
| Claude 3.5 Sonnet | 3.0 | 15.0 | 662.89 | 97.81 | 163.16 |
| Claude 3.5 Haiku | 0.8 | 4.0 | 176.77 | 26.08 | 43.51 |

* Prices are in $\frac{\text{US\$}}{\text{1M Tokens}}$.

$\dagger$ Cost to run all prompting strategies on the given benchmark.

Table 7: Cost estimates for recreating the empirical evaluation with common commercial models.

| Model* | Max Context Length | Max Gen. Tokens | Temperature |
|---|---|---|---|
| meta-llama/Llama-3.1-8B-Instruct | 128000 | 16384 | 0.0 (0.5 for CoT-SC) |
| Qwen/Qwen2.5-14B-Instruct | 128000 | 8192 | 0.0 (0.5 for CoT-SC) |
| Qwen/Qwen2.5-32B-Instruct | 128000 | 8192 | 0.0 (0.5 for CoT-SC) |

* Models were sourced from https://huggingface.co/models.

Table 8: Model configurations used for the empirical evaluation. Additional hyperparameters are found in the accompanying Supplementary Materials.

## C.4 Reproducibility

All results[16], as produced by LM Evaluation Harness, (e.g., LLM inputs and outputs, hyperparameters, runtimes, model configurations, etc.) are found at the following URL: [Link redacted; samples in Supplementary Materials].

All code used to run the evaluations for this paper is found at the following GitHub repository: [Link redacted; code in Supplementary Materials]. Although we used LM Evaluation Harness, we link our fork[17] as significant bug fixes had to be made to get the framework to function as expected. Despite the bugs we encountered, we encourage others to support this open-source project that promotes reproducible results for LLM projects.

## C.5 Configurations

### C.5.1 Models

We include details of model configurations in Table 8. All models were sourced from Hugging Face[18]. Where required, the authors complied with the necessary terms and conditions for gated models.

### C.5.2 Benchmarks

We include the benchmark configurations used for our experiments in Table 9. The underlying datasets for BBH, GSM8K, and MMLU were sourced from Hugging Face[19]. While the fewshot examples for BBH were drawn from the same split used for evaluation, care was taken to ensure that the fewshot examples did not overlap with the target question.

We note in the main text that we selected general-purpose LLM benchmarks for our experiments but recognize that GSM8K could be seen as targeted towards the math domain. While it is true that GSM8K is composed of questions that require basic math, the reasoning capabilities and basic world knowledge it probes are generally applicable. The focus of the benchmark is "properly interpreting a question and reasoning through the steps to solve it" (Cobbe et al., 2021), not to evaluate advanced math skills. We believe this justifies GSM8K's inclusion as a general-purpose benchmark.

---

[16]Provided under the CC-BY-4.0 license.
[17]Under the same MIT license as LM Evaluation Harness.
[18]https://huggingface.co/models
[19]https://huggingface.co/datasets

Table 9: Benchmark configurations used for the empirical evaluation.

| Benchmark | Split | Fewshot Split | # Samples |
|---|---|---|---|
| BBH | test | test* | 6511 |
| GSM8K | test | train | 1319 |
| MMLU | validation | dev | 1531 |

\* Sampled fewshot examples did not overlap with the current benchmark question.

We rely on the steps taken by the authors in creating BBH (Suzgun et al., 2023; Srivastava et al., 2023), GSM8K (Cobbe et al., 2021), and MMLU (Hendrycks et al., 2021) to ensure ethical dataset creation, including the mitigation of bias, offensive content, and personally identifying information. We refer readers to those papers for additional information on the breadth of representation in those benchmarks, such as demographic groups.

### C.5.3 Prompts

To limit the inadvertent effects of linguistic prompt engineering (see Salinas & Morstatter (2024)), we avoided making adjustments to the prompts as they were constructed by default in LM Evaluation Harness, for consistency. As such, we did not include output formatting instructions and used the default prompt template for each model. We excluded output formatting instructions for two reasons: (1) to not interfere with the LLM's natural output for the benchmark question and (2) to allow the benefits of implicit formatting for fewshot strategies. We include an example of a raw text input and output pair for Qwen 2.5 14B on MMLU using Zeroshot CoT below:

**Input**

<|im_start|>system\nThe following are multiple choice questions (with answers) about high school psychology.\n\n<|im_end|>\n<|im_start|>user\nQuestion: Why are psychoanalysts sometimes interested in talking with a patient about dreams?\n(A) Psychoanalysts discovered that neurotransmitter abnormalities sometimes cause dream disturbances.; (B) Since all people are striving for self−actualization, psychoanalysts look at the cognitive obstacles in dreams.; (C) Nonproductive counterconditioning behaviors are sometimes revealed in dreams.; (D) Some psychoanalysts believe that dream symbols represent unconscious conflicts.\nAnswer: Let's think step by step.<|im_end|>\n<|im_start|>assistant\n

**Output**

To determine the correct answer, let's analyze each option in the context of psychoanalytic theory:\n\n(A) Psychoanalysts discovered that neurotransmitter abnormalities sometimes cause dream disturbances.\n− This statement is more aligned with biological psychology rather than psychoanalysis. Psychoanalysis does not focus on neurotransmitters or biological abnormalities as the primary cause of dream disturbances.\n\n(B) Since all people are striving for self−actualization, psychoanalysts look at the cognitive obstacles in dreams.\n− This statement aligns more with humanistic psychology, particularly the work of Carl Rogers and Abraham Maslow, who emphasized self−actualization. Psychoanalysis, as developed by Freud, does not focus on self−actualization or cognitive obstacles in dreams.\n\n(C) Nonproductive counterconditioning behaviors are sometimes revealed in dreams.\n− This statement is more aligned with behaviorism and the concept of counterconditioning, which is not a primary focus of psychoanalysis. Psychoanalysis does not typically discuss counterconditioning in the context of dreams.\n\n(D) Some psychoanalysts believe that dream symbols represent unconscious conflicts.\n− This

Table 10: Licenses for the artifacts used in this work.

| Artifact | License | Notes |
|---|---|---|
| Llama 3.1 8B Instruct | Llama 3.1 Community License | Copyright © Meta Platforms, Inc. |
| Qwen 2.5 14B Instruct | Qwen LICENSE AGREEMENT | Version: September 19, 2024 |
| Qwen 2.5 32B Instruct | Qwen LICENSE AGREEMENT | Version: September 19, 2024 |
| LM Evaluation Harness | MIT License | Copyright (c) 2020 EleutherAI |
| BBH | MIT License | Copyright (c) 2022 suzgunmirac |
| GSM8K | MIT License | Copyright (c) 2021 OpenAI |
| MMLU | MIT License | Copyright (c) 2020 Dan Hendrycks |

Table 11: Average marginal TCs ($\frac{\Delta tokens}{\Delta accuracy}$) calculated between 0 and 3 exemplars and 3 and 8 exemplars for the ablation study on the number of fewshot exemplars.

| Fewshot Range | Marginal TC ($\frac{t}{p}$) | |
|---|---|---|
| | Vanilla Fewshot | Fewshot CoT |
| 0-3 | 117.2 | 30.5 |
| 3-8 | 1621.5 | 553.8 |

```
statement aligns with the psychoanalytic perspective, particularly Freud's
 theory of dreams. Freud believed that dreams are a manifestation of
unconscious desires and conflicts, often symbolized in dream content.\n\
nGiven this analysis, the correct answer is:\n\n(D) Some psychoanalysts
believe that dream symbols represent unconscious conflicts.
```

### C.5.4   Compute

Experiments were run on A100 40GB, A100 80GB, and H100 GPUs, as availability permitted. Configurations were limited to 1 or 2 GPUs per experiment. The experiments were run in tensor parallel or data parallel configurations, depending on the size of the model and the number of GPUs used. The exact configurations per experiment, as well as exact wall times, are detailed in the reproducibility materials referenced in Section C.4. The approximate number of GPU hours across all GPUs was 95.

### C.6   Licensing

We use a number of open-source artifacts in this work. We list the licenses for each in Table 10. We verify that our usage was in accordance with the projects' licenses.

## D   Additional Studies

### D.1   Ablation Study on the Number of Fewshot Exemplars

We present the results from our ablation study on the number of fewshot examples in Figure 3, with detailed results in Table 12. For this experiment, we used the Vanilla Fewshot and Fewshot CoT prompting strategies with the number of exemplars ranging from 0 to 8. As can be seen in Figure 3, the results are noisier but there is a clear trend of diminishing returns as the number of fewshot exemplars increases. To demonstrate this in a way that mitigates the noise, we examine the results piecewise, comparing the average marginal TC between 0 and 3 exemplars and 3 and 8 exemplars. Those values are found in Table 11, for concision. The marginal TC between 0 and 3 fewshot exemplars is, for both Vanilla Fewshot and Fewshot CoT, over an order of magnitude less than between 3 and 8. This indicates a significant decrease in efficiency as token usage increases, which corroborates our observations in Section 4.

Table 12: Detailed results from the ablation study on the number of fewshot exemplars. Token counts represent the total token usage (input and output).

| | # Exemplars | Llama 3.1 8B Instruct | | | Qwen 2.5 14B Instruct | | | Qwen 2.5 32B Instruct | | |
|---|---|---|---|---|---|---|---|---|---|---|
| | | **Tokens** | **Acc.*** | **SE†** | **Tokens** | **Acc.*** | **SE†** | **Tokens** | **Acc.*** | **SE†** |
| Vanilla Fewshot | 0 | 283.8 | 77.6 | 1.15 | 283.0 | 81.7 | 1.06 | 299.4 | 83.7 | 1.02 |
| | 1 | 344.5 | 77.0 | 1.16 | 321.1 | 83.2 | 1.03 | 349.6 | 85.5 | 0.97 |
| | 2 | 401.6 | 77.1 | 1.16 | 386.6 | 83.4 | 1.02 | 418.0 | 85.2 | 0.98 |
| | 3 | 463.4 | 78.1 | 1.14 | 447.1 | 84.2 | 1.00 | 483.1 | 85.1 | 0.98 |
| | 4 | 526.3 | 76.6 | 1.17 | 508.8 | 82.7 | 1.04 | 547.9 | 84.4 | 1.00 |
| | 5 | 607.9 | 78.0 | 1.14 | 574.9 | 83.5 | 1.02 | 613.2 | 84.8 | 0.99 |
| | 6 | 668.8 | 77.6 | 1.15 | 635.1 | 84.2 | 1.01 | 680.4 | 85.0 | 0.98 |
| | 7 | 722.6 | 76.8 | 1.16 | 700.9 | 83.5 | 1.02 | 743.7 | 84.7 | 0.99 |
| | 8 | 787.3 | 79.2 | 1.12 | 768.9 | 83.9 | 1.01 | 810.1 | 84.9 | 0.99 |
| Fewshot CoT | 0 | 273.2 | 78.5 | 1.13 | 281.7 | 81.2 | 1.08 | 294.2 | 83.2 | 1.03 |
| | 1 | 351.3 | 82.2 | 1.05 | 334.1 | 84.5 | 1.00 | 353.7 | 84.3 | 1.00 |
| | 2 | 398.1 | 82.6 | 1.04 | 370.5 | 86.5 | 0.94 | 397.7 | 85.7 | 0.97 |
| | 3 | 462.6 | 82.5 | 1.05 | 418.9 | 88.8 | 0.87 | 443.1 | 87.4 | 0.91 |
| | 4 | 524.8 | 82.0 | 1.06 | 479.6 | 88.4 | 0.88 | 514.2 | 86.8 | 0.93 |
| | 5 | 584.3 | 82.3 | 1.05 | 537.4 | 89.3 | 0.85 | 567.8 | 87.3 | 0.92 |
| | 6 | 664.6 | 81.3 | 1.07 | 615.6 | 88.9 | 0.87 | 638.9 | 88.1 | 0.89 |
| | 7 | 739.2 | 81.4 | 1.07 | 691.0 | 88.9 | 0.87 | 711.0 | 88.4 | 0.88 |
| | 8 | 801.3 | 82.4 | 1.05 | 751.9 | 88.9 | 0.87 | 768.3 | 89.2 | 0.85 |

\* Accuracy as a percentage.
† Standard error.

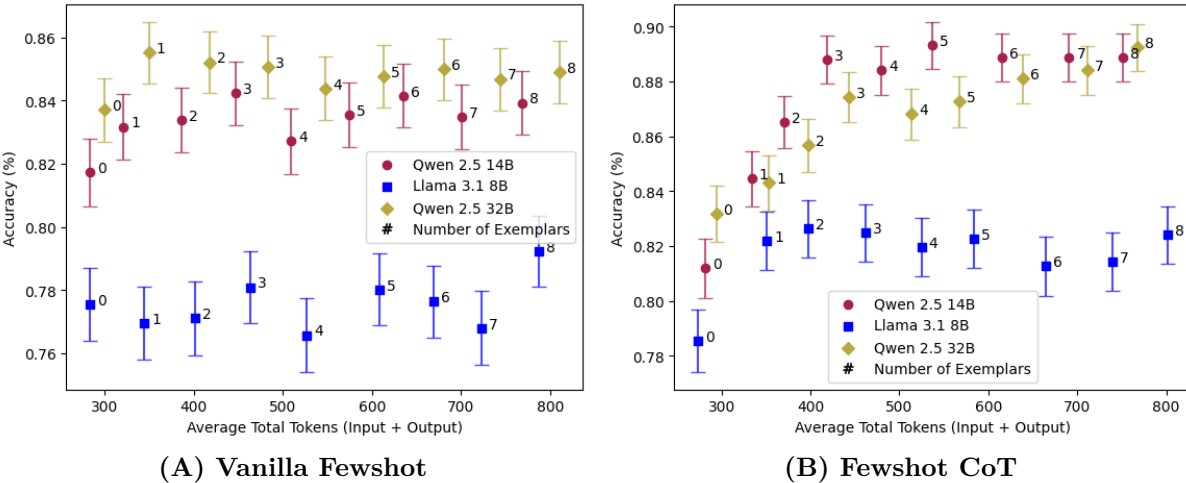

**(A) Vanilla Fewshot**        **(B) Fewshot CoT**

Figure 3: Accuracy and total token usage for the ablation study on the number of fewshot exemplars on the GSM8K benchmark. Standard error bars are included.

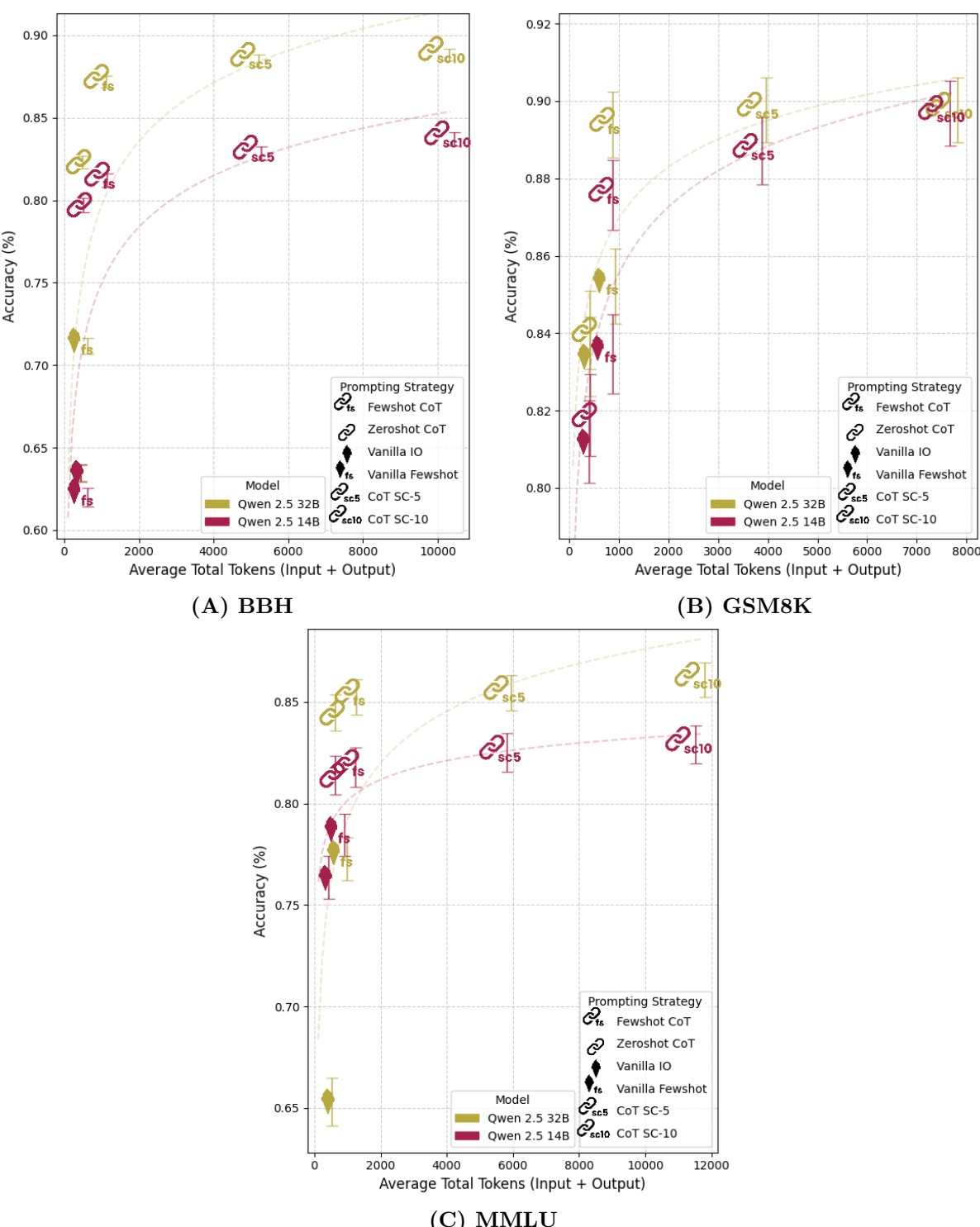

Figure 4: Accuracy and total token usage information for Qwen 2.5 14B and Qwen 2.5 32B from the empirical evaluation. The trend lines reflect the rapid growth of TC for these prompting strategies.

## D.2 Model Size

We included two models from the Qwen 2.5 family to facilitate a discussion on the impact of model size on our observed trends. Selecting two models from the same family ensures that potentially confounding variables, such as differences in training, data collection, and alignment, are presumably kept the same. We present the results from our experiments in Figure 4[20].

We observe that the trend towards diminishing accuracy returns for increased token usage is consistent between the 14B and 32B models. As expected, the 32B model generally outperforms the 14B model. However, we note some instances for prompting strategies that consume fewer tokens, such as Vanilla IO and Fewshot, where the 14B model outperforms the larger one. This suggests that larger models may be able to use additional tokens more effectively, as reflected in the consistently higher accuracy for Fewshot CoT and CoT-SC, but struggle to perform better than smaller models when fewer tokens are provided as context. This provides a promising route for future work.

# E  Use of AI

There was limited use of AI in the research and writing of this work. For writing, ChatGPT[21] was used to rephrase several sentences ($<5$) and to help debug LaTeX and Kubernetes errors. GitHub Copilot[22] was used to help generate some of the plots. Some grammatical suggestions from Writefull's Overleaf integration[23] were considered and included. All AI outputs were thoroughly reviewed by the authors prior to inclusion.

---

[20]These plots are identical to those from Figure 1 but with Llama 3.1 8B excluded, for ease of comparison.
[21]https://chatgpt.com/
[22]https://github.com/features/copilot
[23]https://www.overleaf.com/learn/how-to/Writefull_integration

