# OpenReview forum: "Incorporating Token Usage into Prompting Strategy Evaluation"
_TMLR — Rejected by TMLR_

### Review · Reviewer_Java · 2025-10-11

**Summary Of Contributions:**

### Review Summary

The paper raises an important issue in LLM evaluation,  the computational cost of token usage in prompting strategies. To address this, the authors propose a dual-metric framework: **Big-Otok**, for theoretical token growth analysis, and **Token Cost**, for empirical efficiency measurement. Their main finding is that higher token usage leads to diminishing performance returns.

**Core Problem:** Current evaluation of LLM prompting strategies focuses almost exclusively on task performance (e.g., accuracy), ignoring the computational cost measured in token usage. This can lead to the adoption of inefficient strategies that are effective but wasteful.

**Key Finding:** Through experiments on various models and benchmarks, the paper demonstrates a strong trend of **drastically diminishing performance returns**.

**Audience:**

Yes

**Audience Explanation:**

### **Strengths**

* **Timely and relevant topic:** The study of trade-offs between computational efficiency and benchmark accuracy is both timely and valuable for the field’s progression.

* **Conceptual novelty:** Big-Otok is an interesting adaptation of classical Big-O notation to characterize token growth, while Token Cost provides a simple, interpretable, and practical empirical metric.

* **Empirical consistency:** The observed diminishing returns hold across multiple LLM families (e.g., LLaMA, Qwen), model sizes (8B–32B), and diverse benchmarks (BBH, GSM8K, MMLU), lending credibility to the trend.

**Broader Impact Concerns:**

No broader impact concerns.

**Claims And Evidence:**

No

**Claims Explanation:**

### **Concerns**

**1. Hidden Confounders in Closed-Source LLMs**
The study assumes that user-visible prompts fully determine token usage and performance. This may hold for open-source models but not for closed-source ones (e.g., GPT-4, Claude), which use hidden system prompts and preprocessing layers excluded from token counts. As a result, the reported *Token Cost* may not reflect the true computational input.

**2. Unsubstantiated Causal Claims**
The observed correlation between token usage and diminishing returns is treated as causal without adequate support.

* The experiments do not control for confounders such as task difficulty, exemplar quality, model settings, or output variability.
* Reverse causality remains possible, harder tasks may naturally require longer prompts, making token usage a by-product of task complexity rather than its cause.

**3. Costly Efficiency Evaluation**
Measuring *Token Cost* itself is computationally expensive, requiring evaluations across multiple models, benchmarks, and prompt types, limiting accessibility for smaller research groups.

**Requested Changes:**

**Requested Changes**

The authors should clarify that the relationship between token usage and performance is **correlational rather than causal**, and include **controlled experiments or ablations** to rule out confounders such as task difficulty, model variance, and exemplar quality.

They should also examine the **limitations of Token Cost** when applied to **closed-source LLMs**.

**Suggestions:**

1. Conduct controlled ablations varying token length independently of content and task difficulty.
2. Reframe the conclusion to highlight correlational findings unless causal evidence is provided through additional experiments.

---

### Review · Reviewer_9Vm4 · 2025-10-12

**Summary Of Contributions:**

## Summary
The paper proposes:
- Big-O_tok: a theoretical token-complexity class (constant / linear / polynomial+) for strategies such as 0-shot CoT, few-shot, and self-consistency.
- Token Cost (TC): tokens per percentage-point accuracy, used to compare strategies across BBH, GSM8K, and MMLU on different LLMs.

## Strengths
- Clear articulation of efficiency vs. accuracy with an intuitive TC metric (average and marginal).
- Empirical trends match the claimed Big-O_tok framework.


## Weaknesses
- I think the “theoretical bound” mainly characterizes input-side variables (such as k exemplars or p samples in CoT-SC) rather than output. For models trained for reasoning via RL (e.g. DeepSeek-R1, gpt-5), long reasoning chains can emerge without explicit CoT prompts. In such cases, Big-Otok as presented may misattribute cost because output growth is not modeled separately.
- Scope of models excludes proprietary or reasoning-optimized models; results might differ for models trained to produce long reasoning chains without explicit CoT prompting.


## Suggestions for the authors

1. I think modeling the token usage based on prompting strategy is an interesting and challenging research question. However it requires more thinking if authors want to make a general claim like presented in the paper. The token usage could well be dependent on the task (e.g., simple QA questions vs creative writing), models (e.g., reasoning vs non-reasoning LLMs), not just the prompting strategy.

**Audience:**

No

**Audience Explanation:**

No. The framing is interesting but the contribution lacks sufficient novelty and rigorousness to attract broad attention from TMLR readers.

**Claims And Evidence:**

No

**Claims Explanation:**

No. The proposed framework is overly broad, and key concerns—such as those noted in the weaknesses regarding task dependence, model type, and the mismatch between input- and output-side token growth—are not adequately addressed or supported by evidence.

**Requested Changes:**

See weakness.

---

### Review · Reviewer_XhiG · 2025-10-14

**Summary Of Contributions:**

Summary of contributions:

The paper argues that a trade-off between task accuracy and token use has more practical utility to decide on the prompting strategy than task accuracy alone. The paper introduces a theoretical framework (Big-O-tok) for comparing token usage and an empirical metric Token Cost (TC). The paper shows that increased TC leads to only small gains in accuracy across several standard prompting strategies.

Strengths and weaknesses:

Strengths: The paper tackles a previously overlooked aspect of prompt strategy choice and would be of interest to practitioners.

Weaknesses: the writing isn’t very clear in parts and neither are some aspects of the findings. I detail these below.

**Audience:**

Yes

**Audience Explanation:**

The paper would be of interest to LLM practitioners deciding on a prompting strategy.

**Claims And Evidence:**

Yes

**Claims Explanation:**

While I have some questions about Table 2, overall the paper demonstrates that more sophisticated prompting strategies are associated with a higher TC and smaller accuracy gains.

**Requested Changes:**

Critical:

Results:

1. Could you explain how you envision a practitioner using the theoretical token usage ratios? I’m having difficulty assessing the goodness of fit between the theoretical and observed token usage ratios in Table 2. It seems that the theoretical token usage ratios capture the differences between the classes of prompt strategies while observed usage ratios are gradual; and in some instances the observed token usage ratios are far below those predicted. So I’m not sure how to place this measure in perspective.

Exposition:

1. Some design choices weren’t clear to me until the very end of the paper. For instance, it wasn’t clear what was meant by token efficiency — output, input tokens or both; how the different tokenizers played a role in the token counts; the fact that tokens were approximated through character counts. All of these questions were ultimately addressed at the end of the paper but it would make the paper more reader-friendly if these design choices were addressed earlier in the paper.

Small comments:
1. It would make it easier on the reader if the prompting strategies in Table 1 were labeled with their classes too.
2. Fig. 1 — what do the error bars represent? I also think the y-axis is proportion correct, not %.
3. Could you clarify what the numbers in gray are in Table 2? Averages over all models and datasets? Could you provide some measure of variance like standard error of the mean or 95% CI?

---

### Decision · Action_Editor_NsMo · 2025-11-29

**Recommendation:** Reject

**Audience:**

Yes

**Audience Explanation:**

The topic in this paper is of interest for a large part of the TMLR audience.

**Claims And Evidence:**

No

**Claims Explanation:**

The paper addresses an important problem in LLM evaluation, namely the computational cost of token usage in prompting strategies. Its main finding suggests that increasing token usage yields diminishing performance gains. However, the reviewers raised concerns regarding the reported correlation between token usage and performance gains. While the paper claims that longer prompts lead to diminishing performance returns, reverse causality may still be possible, as more difficult tasks naturally require longer prompts. This points to limitations and insufficiencies in the experimental design to verify the claims, since task difficulty is not explicitly controlled, making token usage potentially a by-product of task complexity rather than its cause. These concerns were not adequately addressed by the authors.